# An Explanatory Model of Emotional Intelligence and Its Association with Stress, Burnout Syndrome, and Non-Verbal Communication in the University Teachers

**DOI:** 10.3390/jcm7120524

**Published:** 2018-12-07

**Authors:** Pilar Puertas-Molero, Félix Zurita-Ortega, Ramón Chacón-Cuberos, Asunción Martínez-Martínez, Manuel Castro-Sánchez, Gabriel González-Valero

**Affiliations:** 1Department of Nursey School, University of Atacama, 1530000 Copiapó, Chile; pilarpuertasmolero@gmail.com; 2Department of Didactics of Musical, Plastic and Corporal Expression, University of Granada, 18071 Granada, Spain; felixzo@ugr.es (F.Z.-O.); rchacon@ugr.es (R.C.-C.); manuelcs@ugr.es (M.C.-S); 3Department of Research Methods and Educational Diagnosis, University of Granada, 18071 Granada, Spain; asuncionmm@ugr.es

**Keywords:** university teachers, emotional intelligence, burnout, stress, nonverbal communication, mental health

## Abstract

The present study set out to define and contrast an explanatory model of perception of stress, the dimensions of burnout syndrome, emotional intelligence, and non-verbal communication in a sample of university teachers. A total of 1316 teachers from Spain, aged between 24 and 70 years (M = 45.64, SD = 10.33) and evenly distributed between both sexes, participated. The measurement instruments employed were the Perceived Stress Scale (PSS), the Maslach Burnout Inventory (MBI), the Trait Meta-Mood Scale (TMMS-24), and the Nonverbal Immediacy Scale (NIS) A structural equation model was produced that demonstrated adequate fit to the empirical data (130,259; df = 9; *p* < 0.001; CFI = 0.907; NIF = 0.914; IFI = 0.923; RMSEA = 0.077). Results revealed that stress relates positively with emotional exhaustion and negatively with personal fulfilment. Emotional exhaustion was associated directly with emotional attention and inversely with emotional clarity and emotional repair, with these being linked to personal fulfilment. Both emotional clarity and repair related positively with non-verbal communication. Conclusions from the present study are that emotional intelligence and body language are two relevant factors in the prevention of burnout syndrome, and as a result can help to ensure the mental wellbeing of university teachers.

## 1. Introduction

Evidence, such as that produced by the International Labour Organisation (ILO), suggests that the working sector is currently finding itself in an alarming situation, in which mental health problems are increasing with 1 in every 10 workers suffering chronic stress, anxiety, burnout, or depression, amongst others. Such issues are the second main cause of unemployment, absence from work, early retirement, and even hospitalization of those afflicted. This has further negative implications with regard to public health, socialization, and economic repercussions [1], such as social isolation, accelerated heart rate, shortness of breath, high blood pressure, sleep disorders, impaired memory, feelings of intense guilt, or unpredictable mood [2,3,4,5].

In this sense, work stress is continuously increasing and reaching towards chronically critical levels [6], which effect cognitive performance and increase susceptibility to diseases [7,8]. Teaching is recognized as one of the professions most affected by the pathologies introduced above. As Horgan et al. [9] notes, 25% of university teachers describe their work as being exhausting or extremely stressful. In periods of stress, the adrenal glands release high levels of cortisol and adrenaline, the latter ensures that blood glucose levels rise significantly in order to provide the body with energy in dangerous situations. If this situation is prolonged in time, it causes corporal and mental exhaustion [10,11,12].

Emotional exhaustion in the university environment is caused by an overload of work tasks, increased responsibility of ensuring a quality education capable of forming professionals of the future, continuous participation in research, enactment of management responsibilities, and continuous contact with students and colleagues [13,14,15,16], which invades and alters the individual’s private life. These factors generate growing psychological pressures that reduce empathy and emotional intelligence, and accentuate cynical attitudes, depersonalization, professional inefficacy, low self-esteem, insomnia, and even eating disorders [17,18,19,20].

Following on from previously discussed research, studies such as those conducted by Fernández-Berrocal et al. [21] and Goleman [22] affirm that emotional intelligence is a psychosocial factor that promotes positive control in situations of conflict. Emotional intelligence is understood as the capacity to perceive and assimilate both personal and detached emotions, enabling them to be adequately managed and permitting the individual to discern between positive and negative aspects so they make correct decisions under pressure [23,24,25]. For this reason, a number of studies have focused on the relationship between this construct and the working environment of teachers as being key to their psychological, physical, and mental wellbeing, as well as augmenting personal satisfaction [26,27,28,29]. This will have a positive impact, both individually and in groups, reducing negative levels of stress.

The emotions experienced by teachers in the lecture hall will determine the way in which they teach. Feelings that have negative connotations influence the capacity for cognitive processing of information, while positive feelings improve creative capacity, which helps individuals to satisfactorily overcome problematic situations [30,31]. This, in turn, favors interpersonal relationships and empathy. Empathy is fundamental in the work environment because it allows us to help and let others help us, reducing the possibility of social isolation [32]. Education professionals who present high perceptions of emotional intelligence are often also characterized as being quick and effective problem solvers, and due to their greater optimism, are capable of generating a more pleasant environment in the lecture hall that connects with their students and livens up the teaching–learning process [33,34,35]. Within the execution of work tasks, there are multiple challenges and adverse situations to be overcome on a daily basis. Adequate use of emotions is critical in these circumstances, enabling generation of whatever means necessary to transform negative feelings and frustration into higher levels of affective commitment [36,37]. Fostering a positive classroom climate contributes to improved personal satisfaction; this fact is essential to ensure the well-being at work [38].

In this sense, efficient use of non-verbal language positively contributes to the development of emotional intelligence in the teaching–learning environment. It assumes a primordial role in the process of teaching and learning, generating feelings of empathy between teacher and student and amongst students themselves. This facilitates social relations, mental wellbeing, motivation, emotional regulation, and academic performance [39,40,41]. It is also an implicit resource to the human being, through which we can understand what other people want to transmit to us. The bad use of the corporal resources causes feelings of rejection on the part of those that surround us, which would enhance burnout, because it produces feelings of incomprehension and low satisfaction by the made work [42].

As has been reported by Kell et al. [43] and Torrents et al. [44], non-verbal aspects play a key role inside the lecture hall, contributing to the building of successful relationships and perceptions of social control. Notwithstanding, non-verbal communication can also manifest negative social consequences, such as the transmission of fragile and unpredictable consequences, when the intention of educators is for their charges to follow a rigid structured process and behavioral pattern. The lack of control over this competence can lead to unexpected negative situations, hence is it crucial that the teacher possesses a good degree of control over their non-verbal communications [45,46]. In this sense, if the teacher does not have control of this capacity, he will lose motivation and self-confidence, which would facilitate the emergence of stress levels and burnout syndrome. This will cause the teacher to want to isolate themselves from his obligations, which could lead to taking time off work [47].

In addition, Mehmet et al. [48] and Valdivieso et al. [49] have remarked that successful work performance of teachers and the teaching–learning process depends to a large extent on the teacher’s ability to react appropriately with improvisations and maintain a close relationship with their students. In this sense, it is necessary to highlight the important role of paralanguage, and specifically, tone of voice, which facilitates the teaching process by catching the attention of students.

Several studies have shown relationships between psychological factors (stress, emotional intelligence, burnout syndrome) and communicative (non-verbal communication). This study investigates whether and to what extent correct emotional management and use of body language are associated with habits of mental well-being in university teachers. The aim of this study was to establish and verify an explanatory model of stress and burnout syndrome in the workplace that considers other potentially influential variables related to physical and mental well-being (emotional intelligence and non-verbal communication).

## 2. Experimental Section

### 2.1. Design and Participants

A descriptive exploratory cross-sectional study was conducted with a homogeneous sample of 1316 university teachers from all public universities of Spain, to the different areas of knowledge and occupational category (Table 1). The sample was made up of 47.3% (*n* = 623) males and 52.7% (*n* = 693) females, with ages between 24 and 70 years of age (Mean (M) = 45.64, standard deviation (SD) = 10.33). Of a total of 99,458 teachers of university academic courses for the 2016/2017 academic year (data extracted from a document produced by the Spanish Ministry of Education, Culture and Sport in 2016) [50], a representative sample of 1316 professionals was derived (with a sampling error of 0.03; confidence interval (CI) = 95.5%) and recruited using stratified random sampling techniques.

### 2.2. Instruments

The Perceived Stress Scale (PSS) is an instrument that was developed by Cohen et al. [51] and adapted into Spanish by Remor [52]. It is composed of 14 items that are rated on a five-point Likert scale, where 0 corresponds to “Never” and 4 corresponds to “Very often”. All items are summed to give a total score with a larger score indicating a greater level of perceived stress. A study conducted by Remor [52] established the reliability of this instrument (determined via Cronbach alpha coefficient analysis), with an *α* = 0.81, where a similar value was detected in the present study, *α* = 0.91.

The Maslach Burnout Inventory (MBI) is an instrument that was developed by Maslach et al. [53], with the Spanish version being validated by Seisdedos [54]. It is composed of a total of 22 items that are responded to on a seven-point Likert scale ranging from 0 = “Never” to 6 = “Everyday”. Burnout syndrome is grouped according to three dimensions, which are: emotional exhaustion (BEE) (items 1, 2, 3, 6, 8, 13, 14, 16, and 20), depersonalization/cynicism (BD) (items 5, 10, 11, 15, and 22), and personal efficacy (BPE) (items 4, 7, 9, 12, 17, 18, 19, and 21). Internal consistency of the scale reported by Seisdedos [54] was *α* = 0.900 for BEE, *α* = 0.790 for BD, and *α* = 0.71 for BPE. In the present study reliability of the scale was *α* = 0.901 for BEE, *α* = 0.74 for BD, and *α*= 0.825 for BPE.

The Trait Meta-Mood Scale (TMMS-24) is an instrument that was developed by Salovey et al. [55], with the Spanish version being validated by Fernández-Berrocal et al. [56]. The instrument is composed of 24 items that are rated on a five-point Likert scale (1 = Disagree; 2 = Slightly agree; 3 = Moderately agree; 4 = Mostly agree; 5 = Totally agree). Emotional intelligence is constituted using three dimensions, which are: emotional attention (EIEA) (items 1, 2, 3, 4, 5, 6, 7, and 8), emotional clarity (EIEC) (items 9, 10, 11, 12, 13, 14, 15, and 16), emotional repair (EIER) (17, 18, 19, 20, 21, 22, 23 and 24). Reliability of the original scale has been reported as *α* = 0.90 for EIEA, *α* = 0.90 for EIEC, and *α* = 0.86 for EIER. In the present study, consistency of the scale was found to be *α* = 0.896 for EIEA, *α* = 0.904 for the category EICE, and *α* = 0.881 for EIER.

The Non-verbal Immediacy Scale, Self-report (NIS), is an instrument that was developed by Richmond et al. [57]. The instrument enables a total score to be produced that reflects the perception an individual has about their use of non-verbal communication. Responses are given to 26 items on a five-point Likert scale, ranging from 1 = “Never” to 5 = “Very often”. Of these items, 13 are positively framed (1, 2, 6, 10, 12, 13, 14, 16, 17, 19, 21, 22, and 25) and 13 are negatively framed (3, 4, 5, 7, 8, 9, 11, 15, 18, 20, 23, 24, and 26). The original version of the scale, obtained an internal reliability value of *α* = 0.80, with a similar value (*α* = 0.828) being produced in the present study.

### 2.3. Procedure

In the first instance, an information letter was sent by the Department of Corporal Expression at the University of Granada to public universities throughout Spain requesting the participation of participants.

Questionnaires were administered to participants following receipt of informed consent. A total of 1403 university teachers agreed to take part in the study and completed questionnaires. Of these, 87 questionnaires had to be excluded due to improper or incomplete completion. Questionnaires were completed via e-mail, which were sent together with instructions on correct completion and a request to complete the questionnaire only once in order to ensure more accurate interpretation of the data. In order to guarantee the reliability of the data, each university teacher was given an identifier code to fill in the questionnaire.

Anonymity and confidentiality of participant data was assured and participation in the study was entirely voluntary. The principles of the ethics committee approved by the University of Granada (462/CEIH/2017) for research set out in the Declaration of Helsinki were followed.

### 2.4. Data Analysis

Data were analyzed using SPPS^®^ version 22.0 (IBM Corp, Armonk, NY, USA) for Windows. Descriptive analysis was performed using frequencies and averages. The program IBM AMOS Graphics version 21.0 (IBM Corp, Armonk, NY, USA) was used to analyze the causal relationship between the variables included in the developed structural equation model. To address study objectives, a path analysis model was defined that included the following endogenous variables: stress, BEE, BD, BPE, EIEA, EIEC, EIER, and overall score of non-verbal communication (NVC).

Model fit was examined with the aim of verifying compatibility of the developed model with the empirical data. Reliability was identified to conform to a goodness of fit criteria [58]. With regard to chi-squared analysis, non-significant values associated with *p* indicate a good model fit. Values for comparative fit index (CFI) analysis are deemed to be acceptable when higher than 0.90 and deemed to be excellent when higher than 0.95. Values for the normative fit index (NFI) should not be higher than 0.90. Values for the incremental fit index (IFI) are deemed to be acceptable when higher than 0.90 and deemed to be excellent when higher than 0.95. Finally, values produced in the analysis of the root mean squared error estimate (RMSEA) are deemed to be excellent when lower than 0.05 and deemed to be acceptable when lower than 0.08 (Figure 1).

## 3. Results

Good model fit was indicated by all indices used to evaluate the fit of the structural equation model. Chi-squared analysis revealed a significant *p*-value (χ^2^ = 130.259; df = 9; *p* < 0.001); however, it must be highlighted that this statistic does not have an upper limit. It, therefore, cannot be interpreted in a standardized way, in addition to being sensitive to sample size, as has been previously discussed. To address these issues, other standardized indices of fit were used, which are less sensitive to sample size. Confirmatory fit index (CFI) analysis produced a value of 0.907, which is acceptable. Normative fit index (NFI) analysis produced a value of 0.914 and the incremental fit index (IFI) value was 0.923, with both being acceptable. Root mean squared error approximation (RMSEA) analysis obtained an acceptable value of 0.077.

Figure 2 displays the estimated values for the measured parameters. For a significant association to be supported, the magnitude of these values must be adequate and the effects significantly different from zero. Improper estimations with negative variances must also be avoided.

Both Table 2 and Figure 2 present the values of the associations between the constructs included in the developed structural equation model. As can be seen, stress was positively and directly associated at the *p* < 0.005 level with emotional exhaustion (*r* = 0.717), and negatively associated with personal efficacy (*r* = −0.491). Statistically significant associations were not found for the relationship between stress and depersonalization/cynicism. Significant associations are observed between emotional exhaustion and depersonalization/cynicism (*p* < 0.005; *r* = 0.541), with these variables being positively related. Depersonalization was also associated with personal efficacy (*p* < 0.01; *r* = −0.129), in this case being negatively related.

In the case of emotional intelligence, a direct association could be seen between emotional attention and emotional clarity (*p* < 0.005; *r* = 0.184), and between emotional repair and emotional clarity (*p* < 0.005; *r* = 0.303), with a stronger correlation being observed with the latter. With regard to the relationship between burnout dimensions and emotional intelligence, statistically significant associations at the level of *p* < 0.005 were seen between emotional exhaustion and emotional attention (*r* = 0.272), between personal efficacy and emotional repair (*r* = 0.449), and between personal efficacy and emotional clarity (*r* = 0.221), with all being positive and direct. Further, statistical associations at the significance level *p* < 0.01 were found in the relationships examined between personal efficacy and emotional attention (*r* = 0.158), and depersonalization/cynicism and emotional clarity (*r* = −0.132), with the former being positive and direct and the latter being inversely related. In the same way, emotional exhaustion demonstrated a negative relationship with emotional repair (*p* < 0.05; *r* = −0.092) and a positive relationship with emotional clarity, although this outcome revealed only a weak correlation strength (*p* < 0.05; *r* = 0.088). No significant differences were observed between depersonalization/cynicism, emotional attention, and emotional repair.

Finally, emotional attention was not found to be associated with non-verbal communication. On the other hand, emotional repair (*p* < 0.005; *r* = 0.249) and emotional clarity (*p* < 0.005; *r* =0.215) were both positively related with non-verbal communication.

## 4. Discussion

The structural equation model reveals the existing causal relationships between stress, the dimensions of burnout syndrome and emotional intelligence, and non-verbal communication within a sample of university teachers. This is a theme that has been studied from different perspectives and in the context of different educational levels. However, no previous studies were found that directly related body language with psychological aspects, such as stress or burnout syndrome, making this a novel aspect of the present study [59,60,61,62,63].

A direct relationship was obtained between stress and emotional exhaustion, while this relationship was inverse in the case of personal efficacy. No relationship was found between stress and depersonalization/cynicism. This outcome may be explained by the fact that subjecting oneself to stress for prolonged periods of time generates a need to distance oneself from one’s profession and feelings of rejection towards the same profession. Such responses are provoked by an excessive workload and the requirement to resolve conflicts that generate emotional and cognitive tensions, bringing with them attentional problems [64,65,66]. In contrast, feelings of satisfaction resulting from the successful execution of one’s work tasks augments motivation and personal wellbeing, and decreases levels of perceived stress [67,68].

With regard to the dimensions of burnout, it was detected that depersonalization relates indirectly to personal efficacy and positively to emotional exhaustion. Previous studies, such as those conducted by Cárdenas et al. [69] and Figueiredo-Ferraz et al. [70], have proposed that the development of cynical attitudes comes from feelings of emotional exhaustion at work, generating attitudes that are defensive and self-protective. A consequence of this is a negative relationship being established between the members who form the educational context.

In consideration of the dimensions of emotional intelligence, all of them show a direct relationship, with the strongest correlation being between emotional repair and emotional clarity [71,72]. Similar findings have been reported previously by Ilaja et al. [73,74], with a sample of 93 university teachers. This can be attributed to the fact that both are strong predictors of this construct, while emotional attention can become detrimental to teachers during certain occasions if it focuses excessively on adverse feelings.

In the case of the relationships given between burnout syndrome and emotional intelligence, a direct association is found between exhaustion and emotional attention, between personal efficacy and emotional repair, and between personal efficacy and emotional clarity. Similarly, to the present study, Augusto-Landa et al. [59] showed that emotional repair related positively and directly with cognitive and emotional coping strategies. Further, personal efficacy and emotional attention where positively associated with each other, while depersonalization/cynicism and emotional clarity were negatively linked in the present study. Augusto-Landa et al. [59] and Llorens et al. [60] have suggested that emotional intelligence acts as a shock absorber for stress that consolidates adaptive coping strategies. In this sense, emotional intelligence serves as a personal resource that facilitates closeness and assists in the avoidance of stress-inducing situations via appropriate emotional utility and management. In the same way, Bisquerra [75] have highlighted emotional education as a tool in response to social needs such as reducing or preventing stress and anxiety. In addition, emotional intelligence has been proposed as an assistive mechanism to successfully cope with the setbacks and work stress faced by teachers in the educational context [76].

In the present study, emotional exhaustion was negatively related with emotional repair and positively related with emotional clarity, although both relationships were found to be weak. In a model proposed previously by Millán-De Lange et al. [61], positive correlations were observed between emotional intelligence and flow disposition, and the possibility of considering these variables as part of one explanatory model was outlined. With this in consideration, although university teachers can reach levels of exhaustion, elements of intrinsic job satisfaction can override and counteract these adverse effects.

Depersonalization/cynicism was not related with either emotional attention or with emotional repair, as was also found in results reported by Mérido-López et al. [77]. In contrast, a study conducted by Balsera-Guera et al. [78] did find the dimension of depersonalization/cynicism to be slightly inversely related with the aforementioned categories of emotional intelligence. As has been shown by Aguayo-Muela [79], cynical attitudes lead individuals to lend their full attention to their strongest negative feelings. Such responses make it even more difficult for teachers to effectively overcome their problems, generating a lack of satisfaction at work and delayed emotional recovery.

Finally, emotional attention was not associated with non-verbal communication in the present study. On the other hand, emotional repair and clarity were positively related with non-verbal communication. This can be explained by the fact that individuals who frequently and adequately use body language, are in turn more competent in understanding and expressing their own feelings, as well as the feelings of others. Further, social relations that encourage conflict resolution in stressful situations are generated more easily through the use of body language [80,81]. It is possible that this construct is influenced by the conscience of oneself, also known as emotional recognition, positively valuing oneself and trusting in oneself [82].

It is important to highlight the main limitations of the present study. The first is linked to its descriptive and cross-sectional design, which only permits interpretation of the considered variables in the context of the defined sample from which they were drawn. A second key limitation was the lack of differentiation with regard to sex amongst the relationships studied. As a result, gender effects relating to the measured constructs could not be examined. Finally, it would have been interesting to extend the spectrum of variables relating to the mental wellbeing of teachers, such as engagement in physical activity, the professional category of teachers, and years of teaching experience.

## 5. Conclusions

The findings reveal that prolonged stress is a predictor of burnout syndrome, finding a relationship between the former and emotional exhaustion. Further, emotional exhaustion was positively associated with emotional attention and was negatively associated with emotional clarity and emotional regulation, with these factors in turn being positively associated with personal efficacy. In the same way, it is highlighted that emotional clarity and regulation is associated with the use of body language. The findings encourage the adequate management and control of emotions as qualities that favor the reduction of stress and prevention of this pathology from initially appearing. 

The findings outline the need to develop programs of physical activity in university teachers, since this reduces the levels of cortisol and norepinephrine, two hormones that accumulate in the body in situations of high stress and anxiety, in addition to releasing endorphin known as the “happiness hormone.” This helps to reduce stress levels and prevent this pathology from appearing. It also boosts emotional intelligence, as physical activity gives them greater ease to deal with their problems quickly and effectively and feelings of empathy. In addition, body communication is improved since the practice of physical activity is elementary to be able to communicate with others. Such abilities can be extrapolated into the work context in the future.

## Figures and Tables

**Figure 1 jcm-07-00524-f001:**
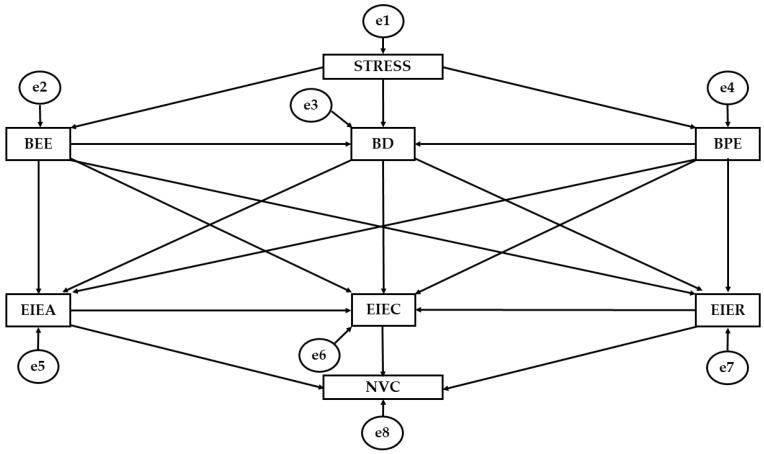
Theoretical structural equation model. Note: BEE, emotional exhaustion; BD, depersonalization/cynicism; BPE, personal efficacy; EIEA, emotional attention; EIEC, emotional clarity; EIER, emotional repair; NVC, non-verbal communication.

**Figure 2 jcm-07-00524-f002:**
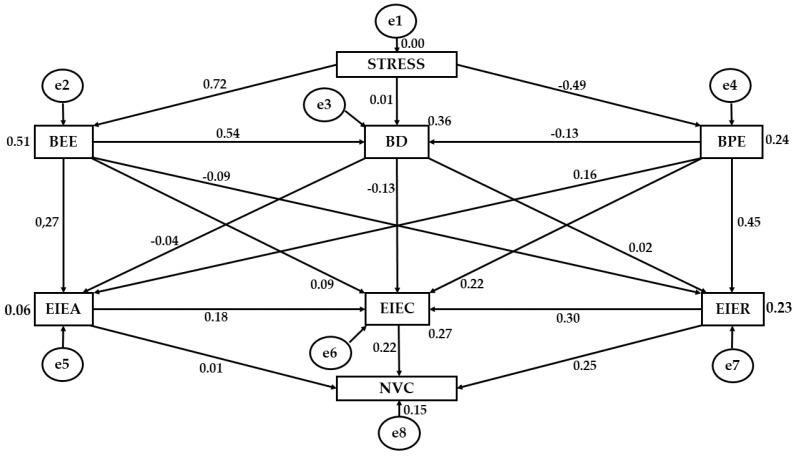
Structural equation model. Note: BEE, emotional exhaustion; BD, depersonalization/cynicism; BPE, personal efficacy; EIEA, emotional attention; EIEC, emotional larity; EIER, emotional repair; NVC, non-verbal communication.

**Table 1 jcm-07-00524-t001:** Demographic characteristics.

**Gender**	
Male	47.3% (*n* = 623)
Famele	52.7% (*n* = 693)
**Age**	
M = 45.64, SD = 10.33	
**Knowledge Areas**	
Social and legal sciences	38.1% (*n* = 502)
Art and humanities	14.7% (*n*=194)
Science	16.7% (*n* = 220)
Health sciences	20% (*n* = 263)
Engineering and architecture	10.4% (*n* = 37)
**Occupational Category**	
Temporary substitute	10.4% (*n* = 137)
University associate	19.1% (*n* = 251)
Contracted doctor	19.6% (*n* = 258)
Assistant doctor	13.3% (*n* = 175)
Holder of university	31.1% (*n* = 409)
University professor	6.5% (*n* = 86)

Note: M, mean; SD, standard deviation.

**Table 2 jcm-07-00524-t002:** Regression weights and standardized regression weights.

Associations between Variables			R.W.				S.R.W.
			EST	S.E.	C.R.	*p*-Value	EST
BPE	←	STRESS	−0.458	0.038	−12.016	***	−0.491
BEE	←	STRESS	0.941	0.043	21.845	***	0.715
BD	←	BEE	0.427	0.042	10.097	***	0.541
BD	←	BPE	−0.144	0.048	−3.004	**	−0.129
BD	←	STRESS	0.006	0.060	0.101	0.919	0.006
EIER	←	BPE	0.511	0.051	10.118	***	0.449
EIER	←	BD	0.025	0.053	0.467	0.641	0.024
EIEA	←	BD	−0.040	0.064	−0.624	0.532	−0.036
EIEA	←	BEE	0.243	0.051	4.724	***	0.272
EIER	←	BEE	−0.075	0.042	−1.775	*	−0.092
EIEA	←	BPE	0.199	0.062	3.222	**	0.158
EIEC	←	EIEA	0.159	0.036	4.434	***	0.184
EIEC	←	EIER	0.291	0.044	6.608	***	0.303
EIEC	←	BEE	0.068	0.041	1.681	*	0.088
EIEC	←	BPE	0.242	0.053	4.564	***	0.221
EIEC	←	BD	−0.130	0.049	−2.629	**	−0.132
NVC	←	EIEA	0.008	0.025	0.312	0.755	0.014
NVC	←	EIER	0.158	0.030	5.242	***	0.249
NVC	←	EIEC	0.143	0.032	4.434	***	0.215

Note 1: R.W., regression weights; S.R.W., standardized regression weights; EST, estimations; S.E., standard error; C.R., critical ratio. Note 2: BEE, emotional exhaustion; BD, depersonalization/cynicism; BPE; personal efficacy; EIEA, emotional attention; EIEC, emotional clarity, EIER, emotional repair; NVC, non-verbal communication; ←, associations between variables. Note 3: * statistically significant association between variables at level 0.05; ** statistically significant association between variables at level 0.01; *** statistically significant association between variables at level 0.005.

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
