# Peer review of "An Explanatory Model of Emotional Intelligence and Its Association with Stress, Burnout Syndrome, and Non-Verbal Communication in the University Teachers"

_jcm, 2018, doi:10.3390/jcm7120524_

Reviewer 1 Report

The presented work is interesting, but I think that it is not well planned or has a clear objective. The objective is simply to see the relationship between variables that we know are related. The model used does not allow knowing the causal directionality of the relationship which would be important and noteworthy in the current issue, that is, only in this case would be novel.

There are four variables and analyzes in a structural model of analysis of the relationship between variables, with a model that does not specify the objective of the analysis, only multiple variables that we know are related and therefore elaborate that model.

I believe that the work is not well planned and the data could offer much more information on the analysis of the problem, which is not known what is, burnout, emotional intelligence, non-verbal communication. It is not known what is intended to be known with the analysis of these variables.

Another of its great limitations would be the selection of the sample and the completion of the questionnaires. It does not make clear in the selection of the sample the qualifications to which the teachers belong, if all the universities are represented, etc. Regarding the obtaining of the data, it is indicated that they have been obtained by email through an online questionnaire. This presents a great limitation, and that is that the veracity of the data is not controlled. It does not indicate if they have asked any control question to know if the answers have not been given at random.

The figures are untranslated. The review made in the introduction is limited, they should focus the analysis on the different variables analyzed, even setting out sections in the introduction, but as it is not clear the objective of the research can not make sections delimiting the purpose of one variable over another, by example, influence and relationship of Emotional Intelligence on burnout in university teaching staff.

I believe that in its current format it does not present a great contribution and highlights the limitations indicated.

Author Response

Dear the editor and reviewers,

We would like to express our gratitude for the time taken to review this manuscript and for the comments made, which we believe to be critical for producing rigorous and quality research. We have detailed below the changes made in the original article: jcm-403395

Modifications have been made in the original manuscript following the reviewers’ comments. For each modification we have written: the original comment as written by the reviewer in addition to the page and line number; and the change made in response to that comment. Changes have been made using the tool “Track changes” enabling editor and reviewers to identify modifications easily.

MODIFICATIONS

EDITOR

Dear editor, first thank you for your suggestions for improving the article.

We have removed the year of publication from the text as you indicated.

In addition, our manuscript has been reviewed by a native scientist.

To make the changes, all the authors have read the text conscientiously and made the changes the reviewers have said.

We remain attentive to your response.

REVIEWER 1

Comment 1:

The presented work is interesting, but I think that it is not well planned or has a clear objective. The objective is simply to see the relationship between variables that we know are related. The model used does not allow knowing the causal directionality of the relationship which would be important and noteworthy in the current issue, that is, only in this case would be novel.

Response 1:

Dear reviewer,

First of all, we would like to thank you for the time spent reviewing the manuscript. We think that once your indications have been made, the quality of the item will improve significantly.

We have modified the objective of the study so that it adjusts more precisely to the results obtained and to the problem that has been analysed (Line 10).

Following the introduction, we have made it by highlighting the alarming situation that currently exists in the field of employment. In the same way, we have researched and searched for information about the factors that generate these pathologies. In addition, there is talk of the benefits and harms of the correct or incorrect use of emotional intelligence, as well as of non-verbal language. The latter plays a fundamental role in the educational work environment.

Comment 2:

There are four variables and analyzes in a structural model of analysis of the relationship between variables, with a model that does not specify the objective of the analysis, only multiple variables that we know are related and therefore elaborate that model.

Response 2:

Thanks for this suggestion of improvement. We have modified the goal so that it agrees with the results and allows us to see the relationships established between the four variables, which are interesting, for example: the direct relationship between exhaustion (burnout syndrome) and emotional attention (emotional intelligence). Or the positive relationships between non-verbal language and emotional intelligence. 

Comment 3:

I believe that the work is not well planned and the data could offer much more information on the analysis of the problem, which is not known what is, burnout, emotional intelligence, non-verbal communication. It is not known what is intended to be known with the analysis of these variables.

Response 3

Dear reviewer, thank you for your contribution to improve and clarify the purpose of this manuscript. To this end, the introduction has been extended so that it is clearer and shows what we are really aiming for by carrying out this study.

Comment 4:

Another of its great limitations would be the selection of the sample and the completion of the questionnaires. It does not make clear in the selection of the sample the qualifications to which the teachers belong, if all the universities are represented, etc. Regarding the obtaining of the data, it is indicated that they have been obtained by email through an online questionnaire. This presents a great limitation, and that is that the veracity of the data is not controlled. It does not indicate if they have asked any control question to know if the answers have not been given at random.

Response 4:

Dear reviewer, thanks to your contribution we have been able to improve and amplify both the sample section and the procedure so that it is adequate.

For the selection of the sample, the collaboration of all the teachers of public universities in Spain was requested, belonging to all areas of knowledge (Social and Legal Sciences, Arts and Humanities, Sciences, Health Sciences and Architecture and Engineering) and occupational categories (Interim Substitute, University Associate, Assistant Doctor, Contracted Doctor, University Holder and Professor). For this purpose, we have included a table with sociodemographic data (Line, 32).

In the procedure, to ensure the veracity and reliability of the data, each subject was assigned an identification code. However, 87 questionnaires were eliminated as they were not well filled out, did not present their assigned identification code, the code was erroneous or the code appeared duplicated (Line, 76).

We hope that with the contributions made, these sections have been corrected.

Comment 5:

The figures are untranslated. The review made in the introduction is limited, they should focus the analysis on the different variables analyzed, even setting out sections in the introduction, but as it is not clear the objective of the research can not make sections delimiting the purpose of one variable over another, by example, influence and relationship of Emotional Intelligence on burnout in university teaching staff.

Response 5:

Dear reviewer, we are sorry about the figures, so thank you again for the time taken to correct this study.  We have done the translation and correction of the figures in the manuscript. In addition, we have modified the objective of the study so that it matches both the results obtained and the introduction (Line 97 and 199).

Comment 6:

I believe that in its current format it does not present a great contribution and highlights the limitations indicated

Response 6:

Dear reviewer, we consider that the study presented shows a contribution to science and would be of great interest to readers, because psychosocial aspects (stress, burnout syndrome and emotional intelligence) are worked on, which today have put society in an alarming situation.

As well as the use of non-verbal language, because it is innate to the human being and its correct use facilitates and contributes to achieving mental well-being. Likewise, after carrying out a bibliographical review we can affirm that it is a subject that has not been studied much, from this perspective.

Therefore, we have tried to include a new variable (non-verbal communication), in order to verify that it favors emotional intelligence and thus mental well-being.

Reviewer 2 Report

I realize that a great work and time has been devoted to this paper. It has a lot of strengths, but I think that some changes should be recommended. 

First, the title does not orient the readers well about the content of the paper. Despite the fact that you cannot affirm any causal relationships, you can suggest it in the title. The title as a list of factors is less motivating. 

Second, the abstract is plenty of acronyms. Please, simplify it for the readers. 

Third, related to the Ethical information, some pieces are missing. For instance, Which is the name of the Ethical committee that approved the study protocol? 

Related to the presentation of the results, I would suggest the authors profit the possibilities that Online journals offer. The figures are little and very difficult to see. Please, try to better design them. 

Fifth, in order to suggest improvements for professors, the authors should include some recommendations and suggestions not only for University's managers, as they do, but also for the own teachers. How could they intervene or manage their own careers and lives in order to avoid burnout?.

Author Response

Dear the editor and reviewers,

We would like to express our gratitude for the time taken to review this manuscript and for the comments made, which we believe to be critical for producing rigorous and quality research. We have detailed below the changes made in the original article: jcm-403395

Modifications have been made in the original manuscript following the reviewers’ comments. For each modification we have written: the original comment as written by the reviewer in addition to the page and line number; and the change made in response to that comment. Changes have been made using the tool “Track changes” enabling editor and reviewers to identify modifications easily.

REVIEW 2

Comment 1:

First, the title does not orient the readers well about the content of the paper. Despite the fact that you cannot affirm any causal relationships, you can suggest it in the title. The title as a list of factors is less motivating. 

Response 1:

Dear reviewer, first of all, we would like to thank you for your time and suggestions for improving the manuscript. We believe that it has been of great relevance.

We have modified the title as indicated. We believe that it now represents the content of the text. Also, if it is still not adequate, we are interested in your suggestion (Line 2).

Comment 2:

Second, the abstract is plenty of acronyms. Please, simplify it for the readers. 

Response 2:

Dear Editor, your suggestions are greatly appreciated as important and help to improve the manuscript. But we have decided not to remove the acronyms from the abstract.

On the one hand, there are the acronyms of the instruments which we believe are necessary.

And on the other hand, the others are the fit indices of the structural equation model and we believe that these should be present, as these are the parameters that indicate that the model has been adjusted correctly.

Comment 3:

Third, related to the Ethical information, some pieces are missing. For instance, Which is the name of the Ethical committee that approved the study protocol? 

Response 3:

Thanks for this suggestion of improvement. The manuscript has been revised and we has included the institution that approved the ethics committee and its identifying code (Line 179).

Comment 4:

Related to the presentation of the results, I would suggest the authors profit the possibilities that Online journals offer. The figures are little and very difficult to see. Please, try to better design them. 

Response 4:

Dear reviewer, thank you for your suggestion because I believe that it improves the study presented. For this reason, we have taken your suggestions on board and the data of the results have been modified in accordance with the regulations of the journal. The figures have also been changed so that they can be displayed correctly.

Comment 5:

Fifth, in order to suggest improvements for professors, the authors should include some recommendations and suggestions not only for University's managers, as they do, but also for the own teachers. How could they intervene or manage their own careers and lives in order to avoid burnout?. 

Response 5:

We appreciate your suggestion, as we believe that this contribution enriches the study.

These modifications have been introduced in the conclusions. We have included strategies so that university professors can manage their own lives and work avoiding chronic stress or exhaustion, through the practice of physical activity, which will favour the emotional intelligence as well as the non-verbal language (Line 338).

Round  2

Reviewer 1 Report

The work has improved, I encourage the authors to carry out studies with fewer variables that analyze the relationship in greater depth.

To be able to accept the article, the authors should review the specialized bibliography on the subject that is currently published in some journals of this platform and that are directly related to this topic. As for example, the monograph that is indicated https://www.mdpi.com/journal/ijerph/special_issues/workplace_health

Author Response

Dear the editor and reviewers,

We would like to express our gratitude for the time taken to review this manuscript and for the comments made, which we believe to be critical for producing rigorous and quality research. We have detailed below the changes made in the original article: jcm-403395

Modifications have been made in the original manuscript following the reviewers’ comments. For each modification we have written: the original comment as written by the reviewer in addition to the page and line number; and the change made in response to that comment. Changes have been made using the tool “Track changes” enabling editor and reviewers to identify modifications easily.

MODIFICATIONS

EDITOR

Dear Editor,

first of all, thank you for your time and dedication in improving the manuscript.

We have made the modification suggested by the reviewer.

We hope that everything is correct and we remain attentive to your response.

REVIEWER 1

Comment 1:

The authors should review the specialized bibliography on the subject that is currently published in some journals of this platform and that are directly related to this topic.

Response 1:

Dear reviewer, we thank you for all the suggestions and contributions that you have told us, as they have been of great relevance for improving the manuscript, and for taking it into account in future publications.

Likewise, we have carried out a bibliographic review as indicated by the journals on this platform, taking those related to our study (Line 345, 353, 369, 380, 399, 488, 490, 498 and 518).

We hope that with this the manuscript is already correct. However, we are attentive to any suggestion that you wish to make to us.

Once again, thank you for taking the time to improve the manuscript.